# Minimizing Dependence between Embedding Dimensions with Adversarial Networks

## Abstract

Learning representations with minimally dependent embedding dimensions can have many potential benefits such as improved generalization and interpretability. This work provides a differentiable and scalable algorithm for dependence minimization, moving beyond existing linear pairwise decorrelation methods. Our algorithm involves an adversarial game where small networks identify dimension relationships, while the main model exploits this information to reduce dependencies. We empirically verify that the algorithm converges. We then explore dependence reduction as a proxy for maximizing information content. We showcase the algorithm's effectiveness on the Clevr-4 dataset, both with and without supervision, and achieve promising results on the ImageNet dataset. Finally, we propose an algorithm modification that gives more control over the level of dependency, sparking a discussion on optimal redundancy levels for specific applications. Although the algorithm performs well on synthetic data, further research is needed to optimize it for tasks such as out-of-distribution detection.

## 1 Introduction

In representation learning (Rumelhart et al., 1986b; Hinton et al., 2006; Bengio et al., 2013), algorithms learn to extract lower-dimensional representations from input data. The quality of representations is typically evaluated by measuring performance in targeted applications. However, the significance of a representation goes beyond enhancing downstream performance: considering properties like fairness, interpretability, and generalization is crucial for any real-world application. Take for instance the application of autonomous driving: it is safety-critical to deploy recognition algorithms that can not only accurately detect samples from the training classes but also identify when a sample comes from an unknown class. Failing to do so may lead the autonomous agent to make poor decisions. We motivate why current recognition algorithms could be inadequate in this scenario with a simple example (illustrated in Figure 1):

**Example 1.** Consider a dataset with images of colored shapes with samples from three training classes: *"red squares"*, *"green triangles"*, and *"blue triangles"*. A classifier could reach a minimum loss value of zero and perfect accuracy by only extracting the color in the output representation. Nevertheless, if one introduced during inference examples from a *"red triangles"* class, the model would predict with high confidence that those examples are *"red squares"*, despite having a shape shared by none of the objects from the *"red squares"* class.

In Example 1, the model relies only on a subset of the features that are relevant to discriminate the training classes, which is problematic when out-of-distribution (OOD, Ben-David et al. (2010); Nguyen et al. (2015)) is required. In this work, we advocate that minimizing the dependence between the embedding dimensions, and thereby minimizing redundancy, could push the model to encode additional features (e.g. both shape and color features in this example) to increase generalization and the robustness of predictions in the presence of OOD samples.

Learning representations with uncorrelated dimensions has a long history in machine learning. Recent methods include applications in self-supervised representation learning (Huang et al., 2018; Zbontar et al., 2021; Ermolov et al., 2021; Bardes et al., 2021) which minimize the pairwise linear correlation between embedding dimensions and an adversarial approach that decorrelates dimensions beyond linearity (Brakel & Bengio, 2017), but that is prone to instability during training.

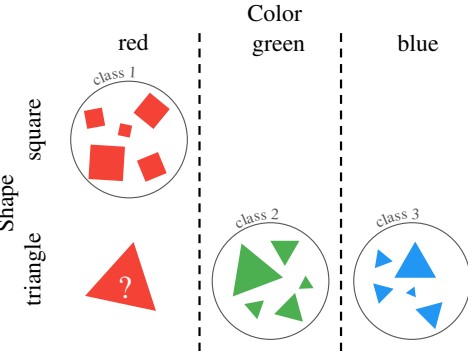

Figure 1: Illustration from Example 1: if a classifier only relied on the color feature, it would predict with high confidence that the red triangle is a red square.

Still, there is to date no stable method for mutual and non-linear dependence reduction. Indeed, finding a training objective for data independence that would be at the same time differentiable and scale to high-dimensional data without making assumptions about the underlying data distributions is notably difficult.

This paper presents a training algorithm to reduce the dependence between learned embedding dimensions using neural networks. The algorithm involves an adversarial game between two types of players: (1) a series of small neural networks are trained to predict one embedding dimension of a representation given the other dimensions, and (2) an encoder is trained to counter reconstruction by updating the representations' distribution.

Experimentally, we show that the game systematically converges to an equilibrium where the dependence between the embedding dimensions is minimal. We then investigate minimizing dependence as a proxy for information maximization and demonstrate the approach's effectiveness both with and without the help of supervision. Specifically, experiments suggest that the algorithm helps a classification model learn concepts beyond label supervision, demonstrating great generalization capabilities. We also show that the method can learn rich representations without supervision by training a self-supervised method on both synthetic data and the large-scale ImageNet dataset.

The main contribution of this paper is to introduce a stable algorithm for nonlinear mutual dependence minimization and to verify its convergence empirically. We further propose a modification of the algorithm that enables networks to keep some level of redundancy when required and study its impact on generalization. Finally, we discuss the implications of the algorithm on information maximization and demonstrate its effectiveness on the Clevr-4 dataset. In particular, the algorithm improves generalization in supervised learning and is also effective without supervision.

## 2 RELATED WORK

Dimensionality reduction has a long history in machine learning, with early works already focusing on finding a representation with uncorrelated variables. Notably, the Principal Component Analysis (PCA, Pearson (1901); Hotelling (1933)) transforms a large set of random variables into a smaller set of uncorrelated variables, known as principal components. This transformation is achieved while preserving the maximal variance in the original dataset, thereby reducing dimensionality without significant loss of information. More recently, approaches in self-supervised representation learning (SSL) also exploited decorrelation (Huang et al., 2018; Zbontar et al., 2021; Ermolov et al., 2021; Bardes et al., 2021) to minimize the dependence between the output dimensions of a deep neural network and as a means to avoid collapsed representations, a common issue in SSL (Jing et al., 2022; Hua et al., 2021). In this work, we investigate decorrelation beyond pairwise linear dependencies.

**Autoencoders.** The PCA algorithm is closely related to autoencoders (AEs, Kramer (1991); Rumelhart et al. (1986a)). An autoencoder consists of two neural networks: an encoder that compresses the input into a latent space and a decoder that reconstructs the input from this representation. Both networks are trained jointly with a reconstruction error. Interestingly, the optimal solution of a

linear AE corresponds to performing PCA. However, unlike PCA, an autoencoder can learn nonlinear dimensionality reductions, but its latent space is not guaranteed to have uncorrelated dimensions. The variational autoencoder (VAE, Kingma (2013)) is an important extension of the autoencoder. It uses a probabilistic approach where the encoder maps the input to a distribution over latent variables and the decoder reconstructs data by sampling from this distribution. Its training involves a criterion minimizing the divergence between the predicted and prior distributions. This extension of AEs allows the generation of new samples similar to the training data. Shortly after its introduction, multiple works (Higgins et al., 2017; Burgess et al., 2018; Kim & Mnih, 2018; Chen et al., 2018) proposed variations to the VAE objective to encourage disentanglement between the latent variables. Intuitively, disentanglement implies learning representations where changes in one factor of variation correspond to changes in a single feature, but disentangled concepts may be dependent. However, while disentanglement is appealing, the problem is ill-defined (Locatello et al., 2019) and is not the focus of this study.

**Input-output mutual information maximization.** Another important line of work on dimensionality reduction relies on the infomax principle (Linsker, 1988; Bell & Sejnowski, 1995), which suggests maximizing the mutual information (MI) between the input data and the output of a neural network to learn informative representations. Similar to our work's objective, MI is an information-theoretic measure of the information shared by two random variables. Mutual Information Neural Estimation (MINE, Belghazi et al. (2018)) provided a first estimate of the MI between high-dimensional continuous random variables using neural networks. Built upon MINE, DeepInfoMax (Hjelm et al., 2019) learns representations by optimizing three criteria: (1) maximizing the MI between the input and the output, (2) maximizing the MI between global and local representations, and (3) matching the output distribution to a uniform prior with the help of adversarial learning. Different from methods derived from the infomax principle, our algorithm minimizes redundancy directly within the representation instead of maximizing a proxy for the input-output MI.

**Adversarial learning.** We now discuss the core training paradigm behind our algorithm and its most notable extensions. Generative Adversarial Networks (GANs, Goodfellow et al. (2014)) are the first approach to train neural networks jointly with an adversarial objective: a generator is trained to create realistic synthetic data, while a discriminator is trained to predict whether a sample came from the training dataset or the generator. Inspired by GANs, Makhzani et al. (2015) introduced Adversarial Autoencoders. This method trains adversarial networks to match the aggregated posterior distribution of an autoencoder's latent space with an arbitrary prior distribution. Thus, sampling from any point of the distribution results in meaningful data generation. InfoGANs (Chen et al., 2016) is another related extension of GANs: they incorporated an information-theoretic criterion into GANs to learn disentangled and interpretable representations. Specifically, the method maximizes a lower bound of the MI between the latent variables and the generated data. Our approach differs from both as it neither matches the representation's distribution to a prior distribution nor maximizes a proxy for MI. Furthermore, our algorithm is not bound to generative networks.

Most similar to our work, Brakel & Bengio (2017) used adversarial networks to decrease dependence by training an encoder to produce samples from a joint distribution that are indistinguishable from samples of the product of its marginals. However, this training objective is unstable and requires careful tuning, while ours systematically converges to the desired equilibrium.

## 3 METHOD

In this section, we first define statistical independence, correlation metrics and motivate the need for a proxy for non-linear dependence reduction. Then, we present a training algorithm to minimize the dependence between embedding dimensions of learned representations.

### 3.1 BACKGROUND AND MOTIVATION

We start by defining independence between two random variables: the continuous random variables $X_1$ and $X_2$ with cumulative distribution functions $F_{X_1}(x_1)$ and $F_{X_2}(x_2)$ are independent if and only if their joint cumulative distribution function $F_{X_1, X_2}(x_1, x_2)$ is equal to the product of their cumulative distribution functions: $F_{X_1, X_2}(x_1, x_2) = F_{X_1}(x_1)F_{X_2}(.x_2)$ for all $x_1$ and $x_2$. Similarly, we define mutual independence for a finite set of random variables $\{X_1, \ldots, X_d\}$: given cumu-

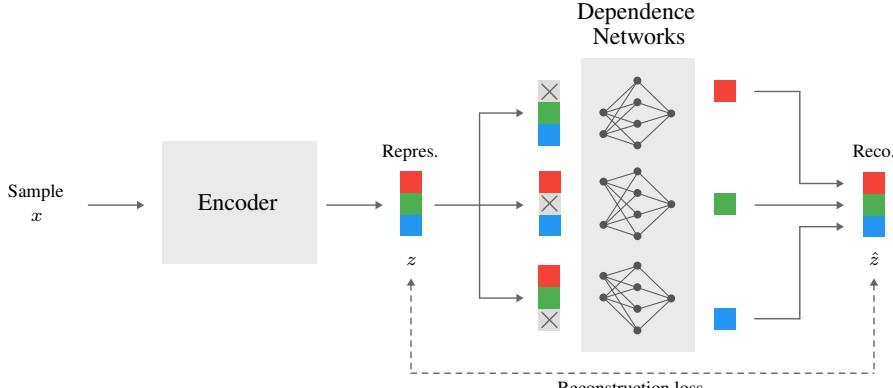

Figure 2: Illustration of the adversarial dependence reduction architecture. The dependence networks minimize the reconstruction error by learning how dimensions relate, while the encoder exploits this information to maximize the error by reducing dependencies.

lative distribution functions $F_{X_1}(x_1), \ldots, F_{X_d}(x_d)$ and the joint cumulative distribution function $F_{X_1, \ldots, X_d}(x_1, \ldots, x_d)$, the random variables are mutually independent if and only if

$$F_{X_1, \ldots, X_d}(x_1, \ldots, x_d) = F_{X_1}(x_1) \cdot \ldots \cdot F_{X_d}(x_d) \text{ for all } x_1, \ldots, x_d \tag{1}$$

We now emphasize an important fact (Driscoll, 1978): mutually independent random variables are also pairwise independent. However, the opposite is not necessarily true — random variables can all be pairwise independent but not mutually independent.

Correlation is a measure of the statistical dependence between two random variables. The term correlation is commonly used in research to refer to Pearson's correlation coefficient, which measures the degree of linear dependence between a pair of random variables. It is defined as the covariance of the two variables normalized by the product of their standard deviations:

$$\rho(X, Y) = \frac{\text{Cov}(X, Y)}{\sqrt{\mathbb{V}(X)\mathbb{V}(Y)}} \tag{2}$$

The Pearson correlation coefficient takes values between zero and one. It is limited to estimating the level of linear dependence between random variables, which means that a zero correlation does not imply independence. This is illustrated in Example 2.

**Example 2.** Let a random variable $X$ be drawn from a uniform distribution in the interval $[-1, 1]$ and $Y = X^2$. The random variables are dependent despite the zero covariance and correlation:

$$\begin{aligned}
\text{Cov}(X, Y) &= \mathbb{E}\left[(X - \mathbb{E}[X])\left(X^2 - \mathbb{E}\left[X^2\right]\right)\right] \\
&= \mathbb{E}\left[X^3\right] - \mathbb{E}\left[X^2\right]\mathbb{E}[X] \\
&= \int_{-1}^{1} \frac{1}{2}x^3 \, \mathrm{d}x - \int_{-1}^{1} \frac{1}{2}x^2 \, \mathrm{d}x \cdot \int_{-1}^{1} \frac{1}{2}x \, \mathrm{d}x = 0
\end{aligned} \tag{3}$$

This result highlights a potential limitation of algorithms relying on Pearson correlation to decrease redundancy: the encoder can minimize the loss with simple non-linearities instead of encoding different concepts. In this work, we take an alternative approach to previous work and present a training algorithm to reduce the dependence between learned embedding dimensions using so-called *adversarial dependence networks*.

Still, a metric is required to estimate our method's decorrelation effect beyond linearity. Distance correlation (Székely et al., 2007) is a non-negative coefficient that characterizes both linear and nonlinear correlations between random vectors. Let $X_1$ and $X_2$ be two random vectors with finite first moments, their respective characteristic functions be denoted $\psi_{X_1}$ and $\psi_{X_2}$, and their joint characteristic function be denoted $\psi_{X_1, X_2}$. Distance covariance measures the distance between their joint characteristic function and the product of the marginal characteristic functions:

$$\mathcal{V}^2(X_1, X_2) = \int_{\mathbb{R}^{p+q}} \left|\psi_{X_1, X_2}(t, s) - \psi_{X_1}(t)\psi_{X_2}(s)\right|^2 w(t, s) dt ds \tag{4}$$

where $w(t,s)$ is a positive weight function and characteristic functions are $\psi_X(t) = \mathbb{E}\left[e^{itX}\right]$. Analogous to Pearson correlation, the squared distance correlation $\mathcal{R}^2$ is defined by: $\mathcal{R}^2(X_1, X_2) = \mathcal{V}^2(X_1, X_2)/\sqrt{\mathcal{V}^2(X_1, X_1)\mathcal{V}^2(X_2, X_2)}$ if $\mathcal{V}^2(X_1, X_1)\mathcal{V}^2(X_2, X_2) > 0$ and 0 otherwise.

Its most significant property is that distance covariance is zero: $\mathcal{V}^2(X_1, X_1) = 0$ if and only if $X_1$ and $X_2$ are independent. Returning to Example 2, we find a non-zero distance correlation between the random variables: $\mathcal{R}^2(X, Y) = 0.5$.

## 3.2 TRAINING ALGORITHM

Consider the representations $\boldsymbol{z}^{(i)} = f_{\boldsymbol{\theta}}(\boldsymbol{x}^{(i)})$ from input samples $\boldsymbol{x}^{(i)}$ of a dataset $\boldsymbol{X} = \{\boldsymbol{x}^{(i)}\}_{i=1}^N$ with i.i.d. samples. The training algorithm involves two types of networks: an encoder $f_{\boldsymbol{\theta}} : \mathcal{X} \to \mathcal{D} \subseteq \mathbb{R}^d$ that learns representations of the training data and a small dependence network for every embedding dimension $g_{\boldsymbol{\phi}_i} : \mathcal{D}_{-i} \subseteq \mathbb{R}^{d-1} \to \mathbb{R}$. The dependence neural networks are trained to learn how dimensions are related. More specifically, every network is given all but one embedding dimension as input and is trained to minimize the mean squared reconstruction error of the missing embedding dimension:

$$\min_{\boldsymbol{\phi}}\|\boldsymbol{z} - \hat{\boldsymbol{z}}\|_2^2 = \min_{\boldsymbol{\phi}} \frac{1}{d} \sum_{i=1}^d \left(z_i - g_{\boldsymbol{\phi}_i}(z_1, z_2, \ldots, z_{i-1}, z_{i+1}, \ldots, z_{d-1}, z_d)\right)^2 \tag{5}$$

We implement dependence networks with Multi-Layer Perceptrons (MLP) since they are universal approximators (Hornik et al., 1989; Cybenko, 1989; Leshno et al., 1993) and can, in theory, approximate arbitrarily well the relation between the variables if given enough capacity.

Intuitively, we now aim to design an algorithm that would exploit the knowledge extracted by the dependence networks to guide the encoder to reduce the dependence between the embedding dimensions. Taking inspiration from Generative Adversarial Networks (Goodfellow et al., 2014), we train both networks simultaneously and model the objective as a two-player zero-sum game where the encoder and dependence networks are respectively trained to maximize and minimize the expected reconstruction error:

$$\min_{\boldsymbol{\phi}} \max_{\boldsymbol{\theta}} \mathbb{E}_{\boldsymbol{z} \in \mathcal{D}(\mathcal{X}, \boldsymbol{\theta})}\|\boldsymbol{z} - \hat{\boldsymbol{z}}\|_2^2 \tag{6}$$

where $\mathcal{D}(\mathcal{X}, \boldsymbol{\theta})$ represents the distribution of representations learned by the encoder, parameterized by $\boldsymbol{\theta}$. The overall architecture is depicted in Figure 2.

**Linear example.** If the dependence networks are implemented with linear layers, each network will learn affine relations between the representation's dimensions to reconstruct the missing embedding accurately. Therefore, training an encoder to counter the reconstruction can be interpreted as a proxy objective for the decorrelation of the dimensions. In particular, we note that a linear dependence network can not succeed when the output dimensions are affinely independent.

**Overcoming trivial solutions.** The encoder is trained to maximize the reconstruction error. It may therefore indefinitely enlarge the norm of the representations to increase the error, even for a constant relative error. We overcome this trivial solution by standardizing the distribution dimension-wise before reconstruction.

$$z_i \leftarrow \frac{z_i - \mathbb{E}[z_i]}{\sqrt{\mathbb{V}[z_i]}} \tag{7}$$

where $\mathbb{E}[z_i]$ and $\mathbb{V}[z_i]$ are respectively the mean and variance of dimension $i$. These quantities are estimated from the current mini-batch and the operation is implemented with a frozen batch-normalization layer following (Ioffe & Szegedy, 2015). Note that the standardization can be applied to the dependence module only and therefore does not limit the modeling capacity of the encoder's output.

**Convergence.** Consider that the dependence networks are trained to reconstruct the standardized representations using a mean squared error loss. When these networks predict the mean vector, the expected value of the loss is, by definition, equal to the variance. For representations standardized to have a zero mean and unit variance, this variance is consistently one. Assuming that the dependence networks can approximate the mean, this forms an upper bound on the expected error since it can be achieved regardless of the standardized distribution of representations. The mean vector is also

the optimal prediction when embedding dimensions are statistically independent since the input of the dependence networks is then irrelevant to the quantities to estimate. Furthermore, a dependence between the representations learned by the encoder could lead to a lower cost loss as the dependence networks may exploit the dependence to improve the reconstruction.

We therefore conjecture that the algorithm with standardized representations converges to a solution where the dependence between the dimensions is minimal and where the dependence networks predict the mean (zero) vector. This convergence is empirically verified in Section 5.1.

**Efficient implementation.** Running many small dependence networks sequentially would be very inefficient for modern GPU architectures. We instead concatenate the inputs from the $d$ networks and implement the dependence networks as one large convolutional network with one-dimensional grouped convolutions (Krizhevsky et al., 2012) with $d$ groups. The grouped convolution effectively isolates the sub-networks from each other while allowing to run all models at once.

## 4 APPLICATIONS

Encoding input data into representations with independent embedding dimensions does not necessarily create a useful structure for downstream tasks. For instance, assume an optimal representation with independent features. Research in nonlinear independent component analysis (NLICA) demonstrated that there are countless ways to transform the representation while maintaining statistical independence (Darmois, 1951; Jutten & Karhunen, 2004). These transformations can involve complex mixing functions that result in representations that may be challenging to interpret or exhibit undesirable properties for certain applications. For examples of mixing transformations, we refer to (Taleb & Jutten, 1999).

Therefore, an additional objective function is required to guide the network towards a practical solution. This work focuses on the application of minimizing dependence as a proxy for information content maximization. We investigate the combination of the adversarial loss with a source of supervision (Section 4.1) and as a self-supervised objective (Section 4.2). In the following, we define the learning objective of the encoder to be a weighted combination of the adversarial loss of maximizing the reconstruction error $\mathcal{L}_{\mathrm{adv}}$ and a task-specific loss function $\mathcal{L}_{\mathrm{task}}$.

### 4.1 INFORMATION MAXIMIZATION FOR CLASSIFICATION

Example 1 illustrated that a classifier could discard relevant features while achieving a global minimum. To counter this limitation, we encourage a model to minimize redundancy as a proxy for maximizing the information in its output representations. Formally, let $z \in \mathbb{R}^d$ be the penultimate representation of a classifier, $z_{\mathrm{std}}$ be its standardized version and $l = Wz + b$ be the logits vector with $W \in \mathbb{R}^{n_c \times d}$ and $b \in \mathbb{R}^{n_c}$ where $n_c$ is the number of training classes. We formulate the loss function as a weighted combination of the adversarial reconstruction loss applied to $z_{\mathrm{std}}$ and the softmax cross-entropy loss:

$$\min_{\boldsymbol{\theta}, W, \boldsymbol{b}} \mathcal{L}_{\mathrm{adv}}(\hat{z}_{\mathrm{std}}, z_{\mathrm{std}}) + \lambda \mathcal{L}_{\mathrm{CE}}(\sigma(Wz + b), y) \qquad (8)$$

where $\sigma$ is the softmax operation.

**Alternative formulation.** In practice, perfectly independent embedding dimensions may not be optimal for classification since it may result in representations that are not linearly separable by the classification head. Hence, we introduce an alternative formulation that only minimizes dependence up to a certain threshold. We formulate the adversarial objective for the encoder as the maximization of a pairwise margin (Cortes, 1995; Tsochantaridis et al., 2005) with parameter $\alpha$:

$$\mathcal{L}_{\mathrm{adv}}(\hat{z}, z) = \max\left(0, \alpha - \|z - \hat{z}\|_1\right) \qquad (9)$$

With this formulation, the encoder does not push the reconstruction error beyond $\alpha$ while dependence networks are still trained to minimize the reconstruction error with no margin: $\|z - \hat{z}\|_1$. The impact of the adversarial objective on decorrelation and generalization is evaluated in Section 5.2 and the two loss formulations are compared in Section 5.3.

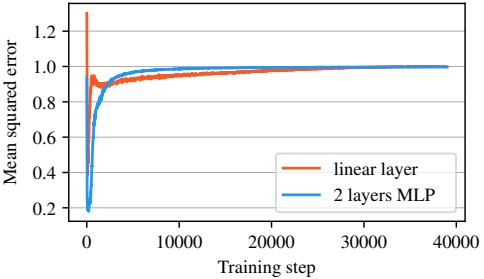 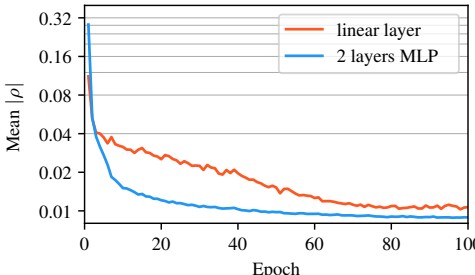

Figure 3: Convergence analysis on TinyImageNet for linear and two-layer dependence networks. **Left:** mean squared error over training, the loss converges to a value of one. **Right:** logarithmic plot of the average absolute value of the Pearson correlation coefficient[1] estimated on the validation set, the value decreases over time.

## 4.2 SELF-SUPERVISED LEARNING

Our algorithm also finds applications in self-supervised representation learning. In a similar spirit to decorrelation SSL techniques (Huang et al., 2018; Zbontar et al., 2021; Ermolov et al., 2021; Bardes et al., 2021), our adversarial objective pushes representations to be minimally redundant and prevents collapse to a trivial solution (Jing et al., 2022). However, different from those approaches, ours is not bound to pairwise linear decorrelation.

We add an invariance loss term that enforces consistency between the input and the output by pushing two augmentations of the same image to be close in the embedding space, i.e. a small change in the input should not lead to a completely different output. Following the methodology from SimCLR (Chen et al., 2020), we sample a minibatch of $n$ images and duplicate every image. We then apply different data augmentations to the two views of each image and enforce invariance by minimizing the MSE between the representations $z$ and $z'$ from corresponding augmented views:

$$\min_{\theta} \mathcal{L}_{\text{adv}}(\hat{z}_{\text{std}}, z_{\text{std}}) + \lambda \|z' - z\|_2^2 \tag{10}$$

In practice, we apply the invariance loss to the standardized representations following the implementation from BYOL (Zbontar et al., 2021).

## 5 EXPERIMENTS

We empirically analyzed the convergence of the adversarial game. Then, we investigated the effect of the training algorithm on information maximization by conducting experiments on the synthetic Clevr-4 dataset (Vaze et al., 2024) in both a supervised and self-supervised setup. In addition, we demonstrated the method's effectiveness on real-world data by applying the approach to SSL on the large-scale ImageNet dataset (Deng et al., 2009).

## 5.1 CONVERGENCE

We analyzed the convergence of the adversarial game combined with a standardization of the representations. Specifically, we trained a ResNet-18 (He et al., 2016) on the TinyImageNet dataset (Le & Yang, 2015) for 100 epochs without data augmentations. Both linear and two-layer dependence models were tested. We additionally trained a ResNet-18 encoder and a two-layer dependence model on the ImageNet dataset for 50 epochs with data augmentations. Detailed hyper-parameters and data augmentations are provided in Appendix C.1.

**Metrics.** We report the average of the absolute value of the Pearson correlation coefficient between all pairs of embedding dimensions over time. Additionally, we estimate non-linear dependences with the distance correlation between one dimension and the random vector composed of the remaining $d - 1$ dimensions. We report the value averaged over the estimates from the $d$ dimensions.

---

[1] The evolution of the Pearson correlation during training is reported instead of distance correlation since distance correlation would be too expensive to estimate at every epoch.

Table 1: Evaluation of the baseline and the adversarial networks on the Clevr-4 dataset. Classification models (CLS) are trained on the *shape* taxonomy.

| method | kNN top-1 accuracy | | | | mean $\mathcal{R}^2$ |
|---|---|---|---|---|---|
| | shape | texture | color | count | |
| CLS baseline | 100.0 | 25.0 | 16.4 | 36.1 | 0.409 |
| CLS adversarial | 100.0 | 83.7 | 100.0 | 39.6 | 0.067 |
| SSL adversarial | 93.8 | 88.5 | 100.0 | 30.6 | 0.081 |

**Results.** The mean squared reconstruction error and the average absolute value of the Pearson correlation coefficient for the TinyImageNet experiments are reported in Figure 3. The linear and non-linear dependence networks converge to low Pearson correlation coefficients of respectively 0.0107 and 0.0088, and the reconstruction error converges to a value of one for both networks. These results support the convergence hypothesis. We further estimate the squared distance correlation: the linear and non-linear variants reach average values of 0.00291 and 0.00057 respectively, which means the approach with a non-linear dependence network reached a five times lower correlation value.

When scaling to the ImageNet dataset, the loss again converges to a value of one and the final squared distance correlation is only 0.00021.

## 5.2 INFORMATION MAXIMIZATION

We investigated the adversarial game as a proxy for information maximization in two different setups. First, we analyzed its effect on the representations when combined with a classification loss. Second, we investigated the self-supervised setup from Section 4.2 under lightweight data augmentations. We trained both approaches on a synthetic dataset for which we have the ground-truth generation factors to ease the estimation of the "informativeness" of the representations.

**Clevr-4 dataset.** The Clevr-4 dataset (Vaze et al., 2024) is an extension of the CLEVR dataset (Johnson et al., 2017). The dataset comprises 100,000 synthetic images representing 3D objects of various shapes, colors, textures, and counts. Each taxonomy has 10 different classes. The label for one taxonomy is sampled uniformly and independently from the other ones, which means that knowing the label for one taxonomy provides no information about the other taxonomies.

**Evaluation protocols.** We investigated generalization capabilities by training a classifier on one taxonomy and evaluating its accuracy on the remaining taxonomies to assess if representations encode features beyond the ones relevant to the training classes. The model is compared with a baseline classifier trained without the adversarial game. We evaluate the accuracy with a simple weighted-nearest neighbor (kNN) classifier trained on top of frozen features following common practice in SSL (Wu et al., 2018; Caron et al., 2021). The kNN algorithm classifies predictions based on the majority class of their nearest neighbors in the embedding space, providing an easy way to assess the clustering quality for every taxonomy. Similarly, the SSL adversarial model is evaluated with a kNN classifier on the four taxonomies.

**Implementation details.** Both the supervised and SSL models are trained on the Clevr-4 dataset for 200 epochs. We apply two data augmentations during training: random horizontal flipping with $p = 0.5$ and random cropping by keeping at least 60% of the image area, followed by resizing to $224 \times 224$ pixels. We train ResNet-18 encoders and two-layer dependence networks. The networks are trained alternately, with one step for each network per iteration. We set the task-specific loss coefficient to $\lambda = 0.2$ in both settings. More details are provided in Appendix C.2. We trained the adversarial networks of the classification and SSL models with respectively an l1 reconstruction loss with a margin of 0.4 and an MSE reconstruction loss on the standardized representations. The different loss formulations are compared in Section 5.3.

**Main results on Clevr-4.** The classification methods are trained on the *shape* taxonomy. The models' accuracy on the validation set and the correlation metrics are reported in Table 1. The embedding dimensions of the baseline are highly correlated, with an average squared distance correlation of 0.409. Furthermore, the performance on the taxonomies for which the model received no supervision is low, which was expected since the model was not incentivized to retain information about the remaining taxonomies. When combining the cross-entropy loss with our adversarial objective, the

correlation drops to 0.067 and the accuracy on the *texture* and *color* taxonomies rises significantly. These results suggest that the adversarial objective reduces redundancy, leading to representations that generalize better. Without the labels from the *shape* taxonomy, the self-supervised model is still able to learn rich representations and reaches a high accuracy on *shape*, *texture* and *color*.

Table 2: kNN evaluation of SSL techniques trained with a ResNet-18 backbone on the Clevr-4 dataset.

| method | kNN top-1 accuracy | | | |
| | shape | texture | color | count |
| --- | --- | --- | --- | --- |
| SimCLR | 58.8 | 50.3 | 91.6 | 28.4 |
| VICReg | 93.1 | 89.2 | 99.5 | 27.5 |
| Ours | 93.8 | 88.5 | 100.0 | 30.6 |

Table 3: Linear evaluation of SSL techniques trained with a ResNet-50 backbone on the ImageNet dataset.

| method | acc. |
| --- | --- |
| MoCo (He et al., 2020) | 60.6 |
| SimCLR (Chen et al., 2020) | 69.3 |
| Barlow Twins (Zbontar et al., 2021) | 73.2 |
| VICReg (Bardes et al., 2021) | 73.2 |
| BYOL (Grill et al., 2020) | 74.3 |
| DINO (Caron et al., 2021) | 75.3 |
| RELICv2 (Tomasev et al., 2022) | 77.1 |
| Ours | 60.6 |

**Comparison to SSL frameworks on Clevr-4.** We compared our method to two popular SSL frameworks on Clevr-4: SimCLR and VICReg. We re-implemented and extensively tuned the two models. We considered variants with and without projection heads. A detailed description of the hyper-parameters tuning is provided in Appendix D and the best-performing models are reported in Table 2. The contrastive method, SimCLR, performs much worse than the decorrelation methods. Both linear decorrelation (VICReg) and non-linear decorrelation (ours) methods achieve similar performance. These results suggest that decorrelation is an effective approach to information maximization and that VICReg does not seem to "cheat with non-linearities" to reduce correlation.

**Model validation on real data.** We then investigated if our method still learns meaningful representations when the training data distribution does not exhibit statistical independence between its underlying concepts[2]. In particular, we trained our SSL technique with a ResNet-50 backbone and a three-layer projection head on the large-scale ImageNet dataset (Deng et al., 2009). We trained two-layer dependence networks on the standardized representations. Performance was then evaluated by training a linear classification head on top of the backbone with frozen weights. The detailed experimental setup is described in Appendix C.3. The main SSL techniques are compared in Table 3. While our approach performs reasonably well, it achieves lower accuracy than the state-of-the-art methods. A possible explanation for the performance gap is that most gains are due to the inductive biases from the loss function that pushes for invariance to the carefully hand-crafted data augmentations used by most techniques. Thus, enforcing strong constraints like mutual independence may force the model to compromise when jointly optimizing both objectives. Nevertheless, this experiment demonstrated that the method leads to useful representations even on real-world data, with an accuracy 606 times higher than random class predictions. For a qualitative evaluation of the predictions, we refer to Appendix B.

### 5.3 Redundancy and downstream performance

What if, for certain applications, the optimal representation required some dependence? For instance in Example 1, we aim to encourage a classifier to retain both the concepts of *color* and *shape*. However, the concepts are not statistically independent. Indeed, assuming a balanced number of samples in every class, we find $P("green" \cap "triangle") = \frac{1}{3}$ and $P("green")P("triangle") = \frac{1}{3}\frac{2}{3} = \frac{2}{9}$, so $P("green" \cap "triangle") \neq P("green")P("triangle")$. Fortunately, the loss function introduced in Section 4.1 provides a remedy to this problem: instead of minimizing dependence to the extreme, the encoder only counters reconstruction until the point where dependence networks are not able to reconstruct samples with an error lower than a margin $\alpha$. Intuitively, this formulation aims to push the network to increase the informativeness of the representations while still allowing for some degree of redundancy.

---

[2]Take for instance the concepts of road and car. While the concepts are distinct, they are likely to often co-appear in the dataset and are therefore correlated in the dataset.

Table 4: Analysis of adversarial loss alternatives on the Clevr-4 dataset. Classification models are trained on the *shape* taxonomy.

| loss | | kNN top-1 accuracy | | | | mean $\mathcal{R}^2$ |
| std. | margin | shape | texture | color | count | |
|---|---|---|---|---|---|---|
| ✓ | ✗ | 100.0 | 21.2 | 16.9 | 22.1 | 0.009 |
| ✗ | 0.4 | 100.0 | 83.7 | 100.0 | 39.6 | 0.068 |
| ✗ | ✗ | 100.0 | 18.4 | 40.4 | 32.2 | 0.287 |

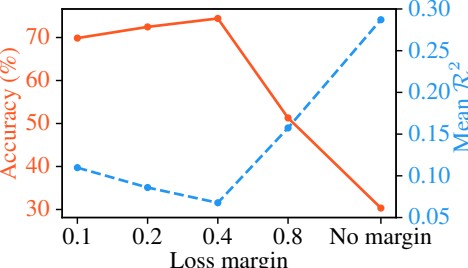

Figure 4: Influence of the loss margin on the correlation and accuracy for classification models trained on the Clevr-4 dataset. The models are trained on the *shape* taxonomy and the kNN accuracy is averaged over the three remaining taxonomies.

To assess the impact of this alternative loss on downstream performance and correlation, we train the classification model from Section 5.2 in three different settings: with standardization of the representations, without standardization but with a margin, and with neither of these. From the results reported in Table 4, it can be observed that the standardized version (first row) achieves by far the lowest correlation level, but its accuracy on the *texture*, *color* and *count* taxonomies is not better than the baseline from Table 1. In contrast, the margin loss (second row) performs well on all taxonomies. Finally, the method with neither a margin nor standardization (third row) performs poorly on the unknown taxonomies and has a much higher correlation level. This can be explained by analyzing that this method indefinitely increased the representation norm instead of decorrelating the variables, reaching an average representation norm of 584.4, while the version with a margin stabilized to around 56.1. We finally report the average accuracy and correlation for different margin values in Figure 4. The figure demonstrates that accuracy increases with the margin up to a margin of 0.4, but that too large margins lead to poor performance on the unknown taxonomies.

## 6    CONCLUSION

In this work, we introduced a representation learning algorithm to minimize the dependence between the embedding dimensions of a representation. Our method involves an adversarial game where small dependence networks identify dimension relationships, while the encoder exploits this information to reduce dependencies. Our problem formulation leads to stable training and empirically converges to minimally dependent representations. Furthermore, we observed that some applications may benefit from representation with a small level of redundancy. Consequently, we introduced an alternative formulation where the encoder only maximizes the reconstruction error of dependence networks up to a set limit. We empirically verified the benefits of our algorithm on the Clevr-4 dataset. Our method significantly improves generalization in supervised learning. It is also effective without supervision, both on synthetic and natural images.

Our study suggests that the optimal level of redundancy varies depending on the application. However, additional research is necessary to gain a deeper understanding of this phenomenon. Another promising area for future work is to explore how our approach's generalization capabilities can be leveraged for out-of-distribution detection or domain adaptation. Lastly, the best architecture for the dependence networks remains to be studied, one may for instance consider Kolmogorov-arnold networks (Liu et al., 2024) for improved interpretability and convergence speed.

## Reproducibility statement

Detailed descriptions of the experimental setups and hyperparameters are available in Appendix C and Appendix D. The adversarial training algorithm is provided in Appendix A. Upon publication of this paper, we will release the full source code and pre-trained model weights in a public repository. This will include a README file with instructions for setting up the environment and reproducing the experiments.

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

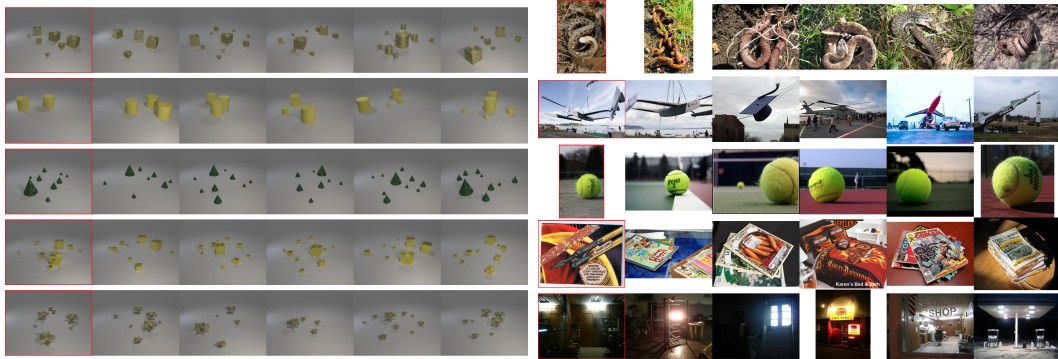

Figure 5: Nearest neighbors visualization for SSL models trained on the Clevr-4 dataset (left) and ImageNet dataset (right). The nearest neighbors visually resemble the query images (highlighted in red).

## A  ALGORITHM

---
**Algorithm 1** Training algorithm for the adversarial dependence minimization
---
**for** number of training iterations **do**
  **for** k steps **do**
    Sample a minibatch of n examples $\{x^{(1)}, x^{(2)}, \ldots, x^{(n)}\}$ from the dataset
    Compute the representations from the encoder $z^{(i)} = f_\theta(x^{(i)})$ for every sample i
    Compute $\mu_j = \frac{1}{n} \sum_{i=1}^n z_j^{(i)}$ and $\sigma_j = \frac{1}{n-1} \sum_{i=1}^n (z_j^{(i)} - \mu_j)^2$ for every dimension j
    Standardize the representations $z_j^{(i)} \leftarrow \frac{z_j^{(i)} - \mu_j}{\sigma_j}$
    Reconstruct the embedding dimensions $\hat{z}_j^{(i)} = g_{\phi_i}(z_1^{(i)} \ldots, z_{j-1}^{(i)}, z_{j+1}^{(i)}, \ldots, z_d^{(i)})$ for every dimension j and every sample i
    Update the reconstruction networks by gradient descent $\nabla_\phi \frac{1}{n} \sum_{i=1}^n \|z^{(i)} - \hat{z}^{(i)}\|_2^2$
  **end for**
  Sample a minibatch of n examples $\{x^{(1)}, x^{(2)}, \ldots, x^{(n)}\}$ from the dataset
  Compute the representations from the encoder $z^{(i)} = f_\theta(x^{(i)})$ for every sample i
  Compute $\mu_k = \frac{1}{n} \sum_{i=1}^n z_j^{(i)}$ and $\sigma_j = \frac{1}{n-1} \sum_{i=1}^n (z_j^{(i)} - \mu_j)^2$ for every dimension j
  Standardize the representations $z_j^{(i)} \leftarrow \frac{z_j^{(i)} - \mu_j}{\sigma_j}$
  Reconstruct the embedding dimensions $\hat{z}_j^{(i)} = g_{\phi_i}(z_1^{(i)} \ldots, z_{j-1}^{(i)}, z_{j+1}^{(i)}, \ldots, z_d^{(i)})$ for every dimension j and every sample i
  Update the encoder by gradient ascent $\nabla_\theta \frac{1}{n} \sum_{i=1}^n \|z^{(i)} - \hat{z}^{(i)}\|_2^2$
**end for**
---

## B  NEAREST NEIGHBORS VISUALIZATION

We visualize the nearest neighbors for the self-supervised models described in Section 5.2. Figure 5 shows the predicted nearest neighbors for five randomly sampled validation images from the Clevr-4 and ImageNet datasets. The left-most image is the query image, and its nearest neighbors are the training samples whose representations have the highest cosine similarity to the query's representation. The figure demonstrates that the nearest neighbors visually resemble the query images on both datasets.

## C    DETAILED EXPERIMENTAL SETUPS

We provide here a detailed description of the training settings and hyper-parameters to facilitate the reproducibility of our experimental results.

The encoder and dependence networks are trained alternately, following the algorithm presented in Appendix A. Epochs are counted relative to the encoder, which means that the dependence networks loop through the dataset $k$ times per encoder epoch.

**Dependence networks.** Dependence networks are always trained with the same optimizer and schedulers as their respective encoder. The default dependence network is a two-layer fully-connected network with a hidden dimension of size 32 and intermediate GELU (Hendrycks & Gimpel, 2016) activation function. There is no activation function at the output of the network.

### C.1    EEXPERIMENTAL SETUP: CONVERGENCE ANALYSIS

**TinyImageNet experiments.** We trained two different dependence networks: a linear and a two-layer fully-connected network. Both are trained on standardized representations. The encoder is a ResNet-18 backbone with no projection head. We used the SGD optimizer with a momentum of 0.9, a learning rate of 0.8, a batch size of 256, and no weight decay. No learning rate schedule is used in this setting. The dependence networks are trained with a ratio of $k = 2$ steps with learning rates of respectively 0.04 and 3.2. The models are trained on the TinyImageNet dataset for 100 epochs without data augmentations, but images are normalized with ImageNet mean and standard deviation per-channel values.

**ImageNet experiment.** We trained two-layer dependence networks on standardized representations. The encoder is a ResNet-18 backbone with a three-layer fully-connected projection head with a hidden dimension of 4096, an output dimension of 512, ReLU activation functions, and intermediate BatchNorm layers. We used the SGD optimizer with a momentum of 0.9, a learning rate of 3.2, a batch size of 1024, and no weight decay. The learning rate follows a cosine decay schedule (Loshchilov & Hutter, 2016) with 10 epochs of linear warmup. The dependence networks are trained with a ratio of $k = 4$ steps with a learning rate of 16. The model is trained on the ImageNet dataset for 50 epochs and follows the same data augmentations as the first views in Grill et al. (2020).

### C.2    EEXPERIMENTAL SETUP: CLEVR-4

**Classification.** The baseline and adversarial approaches are trained with the same set of hyper-parameters, we therefore describe only the adversarial setting. We trained two-layer dependence networks. The encoder is a ResNet-18 backbone with no projection head. We used the SGD optimizer with a momentum of 0.9, a learning rate of 0.1, a batch size of 256, and a weight decay of $2 \cdot 10^{-5}$. The learning rate follows a cosine decay schedule (Loshchilov & Hutter, 2016) with 10 epochs of linear warmup. The dependence networks are trained with a ratio of $k = 1$ steps with a learning rate of 0.3. The adversarial objective is a l1 margin loss on unstandardized representations with margin $\alpha = 0.4$. The task weight is $\lambda = 0.2$. The models are trained for 200 epochs and data augmentations are described in Section 5.2.

**SSL.** We trained two-layer dependence networks on standardized representations. The encoder is a ResNet-18 backbone with no projection head. We used the SGD optimizer with a momentum of 0.9, a learning rate of 0.8, a batch size of 256, and a weight decay of $2 \cdot 10^{-5}$. The learning rate follows a cosine decay schedule (Loshchilov & Hutter, 2016) with 10 epochs of linear warmup. The dependence networks are trained with a ratio of $k = 1$ steps with a learning rate of 0.3. The adversarial objective is a l1 margin loss with margin $\alpha = 0.4$. The task weight is $\lambda = 0.2$. The models are trained for 200 epochs and data augmentations are described in Section 5.2.

### C.3    EEXPERIMENTAL SETUP: IMAGENET SSL

We trained two-layer dependence networks on standardized representations. The encoder is a ResNet-50 backbone with a three-layer fully-connected projection head with a hidden dimension of 4096, an output dimension of 512, ReLU activation functions, and intermediate BatchNorm lay-

ers. We used the LARS optimizer (You et al., 2017) with a momentum of 0.9, a base learning rate of 1.5 with linear scaling rule (Goyal, 2017), a batch size of 1024, and a weight decay of $10^{-4}$. The learning rate follows a cosine decay schedule (Loshchilov & Hutter, 2016) with 10 epochs of linear warmup. The dependence networks are trained with a ratio of $k = 4$ steps with a base learning rate of 6. The model is trained on the ImageNet dataset for 100 epochs and follows the same data augmentations as in Grill et al. (2020).

**Linear evaluation.** We followed standard procedure and trained a linear classifier on top of the frozen representations from the backbone. We used the SGD optimizer with a learning rate of 1.5, a weight decay of $10^{-6}$, a batch size of 256, and trained for 100 epochs. The learning rate follows a cosine decay schedule (Loshchilov & Hutter, 2016). We applied two data augmentations during training: random horizontal flipping with $p = 0.5$ and random cropping by keeping at least 8% of the image area, followed by resizing to $224 \times 224$ pixels. During the evaluation, the images were resized so that the smaller side was 256 pixels wide and then center cropped to $224 \times 224$ pixels.

### C.4 EEXPERIMENTAL SETUP: REDUNDANCY STUDY

The experimental setup is the same as for the classification model from Section 5.2 already described in Appendix C.2. The only difference is that the model trained with standardized representations for reconstruction is trained with a mean squared reconstruction loss. Its dependence networks are trained with a ratio of $k = 2$ steps instead of $k = 1$ since this model did not converge with $k = 1$.

## D CLEVR-4 BASELINES

This section details the hyper-parameter tuning of the SimCLR and VICReg baselines.

We implemented the models following the original papers from SimCLR Chen et al. (2020) and VICReg Bardes et al. (2021). We trained ResNet-18 backbones and trained each model with and without a projection head to find which setup works best for each technique when applied to the Clevr-4 dataset. For a fair comparison, we followed the same experimental setup as for our SSL method: we used the SGD optimizer with a momentum of 0.9, and a weight decay of $2 \cdot 10^{-5}$. The learning rate follows a cosine decay schedule (Loshchilov & Hutter, 2016) with 10 epochs of linear warmup and is scaled with a linear scaling rule (Goyal, 2017).

We ran a grid search on the projection head choice, the learning rate, and the batch size. The models were trained for 80 epochs and the best-performing model was then re-trained for 200 epochs. Its results are reported in Table 2 from Section 5.2.

Results for the grid search on the hyper-parameters from SimCLR and VICReg are reported respectively in Table 5 and in Table 6. We observe that the best-performing model for SimCLR has a projection head, while the VICReg technique works better with no projection head. This observation for VICReg is consistent with findings from our method applied to Clevr-4. This may be because the taxonomies are statistically independent and the augmentations are minimal, reducing the need for a projection head to prevent true invariance to data augmentations (Bordes et al., 2022).

Table 5: Results for the grid-search on SimCLR's hyper-parameters on the Clevr-4 dataset. *LR* stands for base learning rate and *BS* stands for batch size. The best-performing model is **highlighted**.

| head | hyper-parameters output dim. | *LR* | *BS* | kNN top-1 accuracy shape | texture | color | count |
|------|------|------|------|------|------|------|------|
| Identity | (512) | 0.025 | 256 | 42.6 | 40.7 | 83.1 | 24.3 |
| Identity | (512) | 0.025 | 512 | 40.6 | 40.4 | 83.1 | 24.2 |
| Identity | (512) | 0.05 | 256 | 40.7 | 39.1 | 83.9 | 24.1 |
| Identity | (512) | 0.05 | 512 | 47.7 | 39.8 | 80.8 | 24.3 |
| Identity | (512) | 0.1 | 256 | 47.9 | 40.0 | 80.4 | 24.2 |
| Identity | (512) | 0.1 | 512 | 10.1 | 9.9 | 9.8 | 9.4 |
| Identity | (512) | 0.2 | 256 | 39.7 | 34.8 | 76.8 | 24.0 |
| Identity | (512) | 0.2 | 512 | 40.3 | 32.2 | 75.4 | 23.5 |
| Identity | (512) | 0.4 | 256 | 40.9 | 31.6 | 72.0 | 23.6 |
| Identity | (512) | 0.4 | 512 | 42.6 | 29.9 | 73.3 | 22.6 |
| Identity | (512) | 0.6 | 256 | 36.2 | 28.0 | 70.2 | 23.6 |
| Identity | (512) | 0.6 | 512 | 40.6 | 31.9 | 73.4 | 22.8 |
| **MLP** | **128** | **0.025** | **256** | **57.1** | **53.5** | **92.7** | **29.9** |
| MLP | 128 | 0.025 | 512 | 55.6 | 51.0 | 90.9 | 28.2 |
| MLP | 128 | 0.05 | 256 | 57.7 | 51.7 | 92.0 | 28.9 |
| MLP | 128 | 0.05 | 512 | 55.4 | 47.3 | 87.8 | 27.6 |
| MLP | 128 | 0.1 | 256 | 57.6 | 48.4 | 89.1 | 28.0 |
| MLP | 128 | 0.1 | 512 | 47.7 | 41.2 | 86.4 | 27.3 |
| MLP | 128 | 0.2 | 256 | 48.2 | 39.2 | 82.8 | 27.2 |
| MLP | 128 | 0.2 | 512 | 45.9 | 39.3 | 84.2 | 26.5 |
| MLP | 128 | 0.4 | 256 | 44.7 | 34.9 | 74.7 | 26.3 |
| MLP | 128 | 0.4 | 512 | 46.0 | 35.6 | 75.1 | 26.7 |
| MLP | 128 | 0.6 | 256 | 44.0 | 34.4 | 73.8 | 26.6 |
| MLP | 128 | 0.6 | 512 | 40.8 | 37.3 | 70.4 | 26.4 |
| MLP | 512 | 0.025 | 256 | 47.3 | 47.7 | 89.1 | 29.4 |
| MLP | 512 | 0.025 | 512 | 46.7 | 46.2 | 89.4 | 28.4 |
| MLP | 512 | 0.05 | 256 | 46.6 | 45.7 | 88.0 | 28.3 |
| MLP | 512 | 0.05 | 512 | 46.2 | 43.7 | 87.1 | 28.2 |
| MLP | 512 | 0.1 | 256 | 47.0 | 42.6 | 84.9 | 28.2 |
| MLP | 512 | 0.1 | 512 | 54.4 | 44.2 | 85.7 | 27.3 |
| MLP | 512 | 0.2 | 256 | 44.8 | 37.9 | 81.1 | 27.1 |
| MLP | 512 | 0.2 | 512 | 49.4 | 42.6 | 79.6 | 26.3 |
| MLP | 512 | 0.4 | 256 | 44.9 | 38.7 | 75.7 | 26.4 |
| MLP | 512 | 0.4 | 512 | 46.6 | 37.0 | 75.9 | 26.3 |
| MLP | 512 | 0.6 | 256 | 44.4 | 35.0 | 75.1 | 27.4 |
| MLP | 512 | 0.6 | 512 | 45.2 | 34.1 | 73.3 | 27.0 |

Table 6: Results for the grid-search on VICReg's hyper-parameters on the Clevr-4 dataset. *LR* stands for base learning rate and *BS* stands for batch size. The best-performing model is **highlighted**.

| | hyper-parameters | | | kNN top-1 accuracy | | | |
| head | output dim. | *LR* | *BS* | shape | texture | color | count |
|---|---|---|---|---|---|---|---|
| Identity | (512) | 0.005 | 256 | 82.1 | 86.7 | 100.0 | 28.2 |
| Identity | (512) | 0.005 | 512 | 88.3 | 87.4 | 100.0 | 26.5 |
| Identity | (512) | 0.01 | 256 | 86.9 | 85.3 | 100.0 | 34.1 |
| **Identity** | **(512)** | **0.01** | **512** | **91.4** | **88.6** | **100.0** | **27.2** |
| Identity | (512) | 0.025 | 256 | 84.1 | 79.4 | 99.5 | 29.8 |
| Identity | (512) | 0.025 | 512 | 86.2 | 84.3 | 99.4 | 31.9 |
| Identity | (512) | 0.05 | 256 | 73.9 | 71.0 | 98.9 | 26.3 |
| Identity | (512) | 0.05 | 512 | 81.0 | 77.5 | 98.9 | 28.8 |
| Identity | (512) | 0.1 | 256 | 72.5 | 66.9 | 98.8 | 24.2 |
| Identity | (512) | 0.1 | 512 | 60.6 | 65.0 | 98.4 | 23.7 |
| Identity | (512) | 0.2 | 256 | 63.1 | 57.0 | 98.2 | 22.9 |
| Identity | (512) | 0.2 | 512 | 51.1 | 54.8 | 97.0 | 20.0 |
| Identity | (512) | 0.4 | 256 | 44.4 | 48.3 | 97.0 | 22.9 |
| Identity | (512) | 0.4 | 512 | 50.3 | 51.2 | 97.4 | 23.5 |
| MLP | 128 | 0.005 | 256 | 51.0 | 57.6 | 99.5 | 30.4 |
| MLP | 128 | 0.005 | 512 | 51.4 | 61.3 | 99.0 | 29.7 |
| MLP | 128 | 0.01 | 256 | 44.4 | 54.9 | 98.2 | 29.0 |
| MLP | 128 | 0.01 | 512 | 47.7 | 57.5 | 98.3 | 27.7 |
| MLP | 128 | 0.025 | 256 | 34.8 | 51.1 | 97.0 | 24.2 |
| MLP | 128 | 0.025 | 512 | 42.8 | 50.6 | 97.0 | 25.8 |
| MLP | 128 | 0.05 | 256 | 36.1 | 44.6 | 95.8 | 25.0 |
| MLP | 128 | 0.05 | 512 | 36.8 | 49.9 | 97.3 | 24.7 |
| MLP | 128 | 0.1 | 256 | 31.1 | 42.3 | 95.3 | 24.5 |
| MLP | 128 | 0.1 | 512 | 34.3 | 44.3 | 96.2 | 25.2 |
| MLP | 128 | 0.2 | 256 | 30.3 | 40.9 | 95.8 | 23.3 |
| MLP | 128 | 0.2 | 512 | 31.7 | 42.7 | 96.3 | 23.5 |
| MLP | 128 | 0.4 | 256 | 28.1 | 25.8 | 94.5 | 23.2 |
| MLP | 128 | 0.4 | 512 | 26.8 | 14.1 | 33.3 | 21.5 |
| MLP | 512 | 0.005 | 256 | 63.7 | 68.3 | 99.5 | 30.5 |
| MLP | 512 | 0.005 | 512 | 63.8 | 66.1 | 99.4 | 28.4 |
| MLP | 512 | 0.01 | 256 | 59.6 | 63.5 | 98.7 | 28.3 |
| MLP | 512 | 0.01 | 512 | 61.5 | 64.3 | 98.6 | 26.2 |
| MLP | 512 | 0.025 | 256 | 61.2 | 60.2 | 97.9 | 26.6 |
| MLP | 512 | 0.025 | 512 | 62.2 | 62.8 | 98.0 | 24.7 |
| MLP | 512 | 0.05 | 256 | 59.8 | 58.5 | 97.4 | 25.7 |
| MLP | 512 | 0.05 | 512 | 58.6 | 59.1 | 97.2 | 23.9 |
| MLP | 512 | 0.1 | 256 | 57.2 | 57.2 | 97.1 | 24.7 |
| MLP | 512 | 0.1 | 512 | 57.2 | 56.8 | 97.0 | 23.5 |
| MLP | 512 | 0.2 | 256 | 56.9 | 52.9 | 96.6 | 24.5 |
| MLP | 512 | 0.2 | 512 | 55.7 | 54.2 | 96.8 | 23.5 |
| MLP | 512 | 0.4 | 256 | 46.2 | 49.0 | 96.3 | 23.7 |
| MLP | 512 | 0.4 | 512 | 44.1 | 51.2 | 96.5 | 23.0 |

