# OpenReview forum: "Minimizing Dependence between Embedding Dimensions with Adversarial Networks"
_ICLR.cc/2025/Conference — ICLR 2025 Conference Withdrawn Submission_

### Official Review · Reviewer_vQQ7 · 2024-10-31

**Soundness:** 2
**Presentation:** 3
**Contribution:** 2
**Rating:** 5
**Confidence:** 3

**Summary:**

This paper introduces a differentiable and scalable algorithm for dependence minimizations in representation learning. The core idea is to employ an adversarial game where small neural networks identify dimension relationships, while the main model (encoder) exploits this information to reduce dependencies. The method is tested on synthetic datasets, showing improved generalization and interpretability of the learned representations.

**Strengths:**

1.	The use of adversarial training to minimize dimension dependence is a novel solution that moves beyond traditional linear decorrelation methods.
2.	The algorithm is shown to be effective in both supervised and self-supervised learning scenarios.

**Weaknesses:**

1.	There is a lack of solid theoretical underpinnings for the proposed method to demonstrate convergence and the effectiveness of the algorithm.
2.	While the paper claims stability, the adversarial approach might introduce additional complexity and potential instability in training.
3.	The experimental evaluations on real-world datasets are quite limited. A comprehensive empirical study over different models and datasets would be beneficial.

**Questions:**

1.	How does the algorithm scale to larger and more complex datasets, especially those with high-dimensional embeddings?
2.	What are the computational costs associated with training the additional dependence networks, and how do they compare to standard representation learning methods?

---

### Official Review · Reviewer_1pgN · 2024-11-03

**Soundness:** 3
**Presentation:** 2
**Contribution:** 2
**Rating:** 5
**Confidence:** 3

**Summary:**

In this work, the authors introduce a novel methodology for learning representations in which the dimensions are optimized to minimize their mutual dependence. To achieve this, the authors introduce an adversarial framework in which, for a given latent representation, a set of dependence neural networks attempt to reconstruct each dimension of the representation using all other dimensions. The latent representation is then optimized to maximize the reconstruction error produced by the dependence neural networks, thereby encouraging independence among dimensions. Formally, the problem introduced in this work is formulated as a minimax optimization problem of the reconstruction error, and to avoid trivial solutions, the authors propose to measure the reconstruction error of the dependence networks on the standardized representations. The authors also propose a relaxed formulation of their problem, where the reconstruction error is truncated up to a certain level $\alpha$, and which is implemented using the Hinge loss on the reconstruction error. Additionally, the authors present two applications of their approach when combined with either a supervised loss or a self-supervised metric. Finally, the authors evaluate experimentally their approach to show both the convergence and the performances obtained on one synthetic and two real-world problems. More specifically, they first show that the proposed adversarial algorithm converges towards an uncorrelated representation. Then they compare their approach for classification on the Clevr-4 dataset against the baseline where no adversarial training is considered and show that the representation obtained with the adversarial training can generalize to new classification tasks on the same data. The authors also compare their approach with SoTA methods in self-supervised learning (SSL), demonstrating comparable performance on out-of-distribution classification tasks on the Clevr-4 dataset. The authors also show that their approach underperforms relative to SoTA methods on the ImageNet dataset.

**Strengths:**

The proposed approach is, to the best of my knowledge, novel and exemplifies another instance where adversarial training proves beneficial. The method presented here appears to circumvent common limitations associated with adversarial techniques, as their experimental results suggest a well-behaved convergence of the algorithm. Furthermore, the authors provide a thorough explanation of related work, clearly positioning their contributions. They also emphasize simple yet illustrative examples to motivate their approach, enhancing the readability and accessibility of the paper.

**Weaknesses:**

The paper presents several issues regarding the presentation of the experimental results, as well as some inconsistencies in the methodology presented. Additionally, it remains unclear from the experiments what advantages this approach offers over more standard methods that penalize only linear correlations.

Concerning the setting, the authors formulate a minimax problem in Equation (6); however, in practice, it seems that they apply a maximin formulation to extend their approach to different applications such as classification or SSL. Furthermore, distance correlation is referenced only briefly in Section 5.1 while it is extensively presented in the background section, and the reported values lack context, as they are not compared against other methods, leaving their significance ambiguous. In Section 5.2, the explanation of different settings within the same section introduces confusion. For improved readability, it would be beneficial to separate these settings into distinct sections. Multiple metrics are also used throughout the experiments—one for the encoder, another for the dependence networks, and a third for specific applications, and it is unclear which metric applies to each experimental setup. Clarification on this point would help improve the clarity of the experimental analysis. Finally, VICReg achieves comparable performance on the Clevr-4 dataset while outperforming the proposed method on ImageNet, despite relying solely on enforcing decorrelation in the representation. Overall, while the proposed approach appears to be novel, the advantages it offers over other baseline methods could benefit from further justification and clarification.

typo l 193: The Pearson correlation coefficient takes values between "minus one" and one.

**Questions:**

Why not directly learn a standardized representation?
How would you justify the advantage of the proposed approach over VICReg?
How does the proposed method with linear dependence networks compare to VICReg?

**Details Of Ethics Concerns:**

N.A.

---

### Official Review · Reviewer_gdUp · 2024-11-07

**Soundness:** 2
**Presentation:** 3
**Contribution:** 1
**Rating:** 3
**Confidence:** 3

**Summary:**

The paper proposes a new learning objective that augments the standard (self)-supervised learning loss with a regularization loss. The latter minimizes the dependence among all embedding dimensions by adversarially training d networks that measure the dependence between each dimension and all the others. The new objective can be applied to improving information maximization for representation learning and also self-supervised learning to prevent the trivial solutions. The model is applied to various experimental tasks and compared with multiple baseline methods.

**Strengths:**

1. The paper is well written and easy to follow

2. The proposed method is clearly presented and has applicability to different learning problems

**Weaknesses:**

1. The proposed method lacks enough justification on why the adversarial training networks can minimize the dependence among embedding dimensions. How do the local optima of the objective impact the learning? Why minimizing the dependence among embedding dimensions is beneficial for learning? The authors need to supplement more evidence and examples for justifying the model

2. The advantages of the proposed objective is not clear and ungrounded. Sec 4 is entirely not clear. In Sec. 4.1, it is unjustified why the proposed objective can maximize the information of learned representations. In Sec. 4.2, it is unclear how the objective can prevent the degenerate solution.

3. From the experimental results reported in Sec. 5, there is no clear improvement over the baselines, which questions the effectiveness of the new method.

**Questions:**

1. Why the adversarial training networks can minimize the dependence among embedding dimensions.

2. How do the local optima of the objective impact the learning?

3. Why minimizing the dependence among embedding dimensions is beneficial for learning? The authors need to supplement more evidence and examples for justifying the model.

4. Why the proposed objective can maximize the information of learned representations? How the objective can prevent the degenerate solution for self-supervised learning?

---

### Official Review · Reviewer_ar1H · 2024-11-08

**Soundness:** 2
**Presentation:** 2
**Contribution:** 2
**Rating:** 5
**Confidence:** 4

**Summary:**

The paper presents an adversarial training algorithm aimed at reducing dependencies among embedding dimensions in neural networks, thereby improving representation, generalization, and interpretability. This method consists of two main components: small dependence networks that forecast individual embedding dimensions from the others and an encoder that adjusts the embeddings to lessen these dependencies by maximizing the prediction error. As a result, the approach creates a minimally dependent embedding space, enhancing model robustness and decreasing redundancy.

Key contributions include:
1) A stable adversarial approach for nonlinear mutual dependence minimization extends beyond linear pairwise decorrelation.
2) An adjustable redundancy control mechanism that allows for some dependencies when beneficial, providing flexibility for various applications.
3) Empirical validation across synthetic and real datasets (Clevr-4 and ImageNet) demonstrates improvements in generalization and information retention without supervision.

This work offers a scalable, differentiable solution for dependence minimization in representation learning, opening the door to applications such as out-of-distribution detection and self-supervised learning.

**Strengths:**

Originality

Novel Adversarial Approach: The adversarial training method for decorrelating embeddings, moving beyond linear and pairwise decorrelation, is indeed innovative. However, while the approach is novel, it heavily draws on existing concepts from adversarial representation learning, such as InfoGAN (Chen et al., 2016) and Mutual Information Neural Estimation (MINE) (Belghazi et al., 2018). Thus, the originality of this paper lies in adapting these ideas for decorrelation rather than proposing a fundamentally new paradigm.

Controlled Redundancy: The redundancy control is a practical addition, offering the flexibility to adjust embedding independence levels to suit specific applications. However, the paper does not delve deeply into how to optimally balance this redundancy, which is crucial, as excessive independence may harm task-specific performance. Exploring methods to dynamically balance independence and redundancy could make this contribution even more impactful.

Quality

Strong Empirical Evaluation with Some Limitations: The paper provides comprehensive experiments on synthetic and real datasets that support claims about decorrelation, generalization, and robustness. That said, the results on ImageNet fall short compared to state-of-the-art (SOTA) self-supervised methods like BYOL (Grill et al., 2020) and Barlow Twins (Zbontar et al., 2021). The weaker ImageNet performance suggests potential limitations in scaling or generalizing this approach to complex data. Addressing these limitations by incorporating stronger augmentation strategies or alternative evaluation tasks, such as object detection, could help clarify the practical value of the approach.

Theory and Empirical Validation: While the theoretical discussion around stability and convergence is thorough, it lacks a rigorous exploration of how architectural choices (e.g., depth and regularization of dependence networks) impact stability across datasets. Additionally, the empirical validation would be stronger if the paper included sensitivity analyses to illustrate robustness across diverse architectures or hyperparameter choices.

Clarity
Clear Presentation but Dense: The paper is well-structured, but certain sections, particularly those explaining the adversarial setup, are dense and could benefit from additional high-level summaries. For instance, the technical sections on dependence networks and standardization processes may be challenging for readers less familiar with adversarial learning frameworks. Including intuitive explanations or simplified diagrams would enhance clarity and accessibility.

Limited Background Context for Novelty: Although the authors describe their contributions relative to prior methods, the paper could more clearly contrast its method against SOTA techniques in decorrelation, such as Whitening-MSE (Hua et al., 2021) and VICReg (Bardes et al., 2021). A deeper discussion on where this approach significantly diverges from or extends previous work would make the novelty more explicit.

Significance

Impact on Representation Learning: The approach targets a meaningful problem by aiming to improve generalization through reduced redundancy in representations, which is beneficial for out-of-distribution (OOD) detection and self-supervised learning (SSL). However, the broader impact of this work is somewhat unclear, as the performance lag on real-world data suggests limitations in highly complex tasks, where robustness and generalization are most critical.

Applicability in Other Domains: While the paper demonstrates effectiveness in image data, it doesn’t evaluate the approach in multimodal or textual data, where decorrelation could also be valuable. Testing this approach on tasks beyond image classification, such as speech recognition or multimodal embeddings, could substantiate its relevance to the broader ICLR community and indicate its versatility across domains.

**Weaknesses:**

Comparative Analysis with State-of-the-Art (SOTA):

Weakness: Although the paper compares its approach to well-known methods like SimCLR and VICReg, it lacks a more comprehensive benchmark against other recent advancements in self-supervised decorrelation and redundancy reduction, such as Barlow Twins (Zbontar et al., 2021), VICReg (Bardes et al., 2021), and Whitening-MSE (Hua et al., 2021). Each of these methods focuses on decorrelating embedding dimensions to varying extents, and their performance is well established across multiple datasets.6

Suggestion: Expanding the experimental comparisons to include these recent methods would strengthen the claim of decorrelation efficacy and generalization improvements. For example, Barlow Twins uses a cross-correlation loss to enforce independence; VICReg introduces variance-covariance regularization, and Whitening-MSE directly whitens embeddings to reduce redundancy. Benchmarking the proposed method against these approaches—especially on complex datasets like ImageNet—would provide a clearer context for its performance and practical advantages.

Reliance on Architectural Choices:

Weakness: The paper’s approach depends on specific architectural and tuning choices, especially regarding dependence networks, which may limit its generalizability across diverse contexts. The paper mentions that a two-layer network structure and specific regularization techniques are required to avoid trivial solutions, suggesting sensitivity to hyperparameters.

Suggestion: A detailed analysis or ablation study of different architectural choices for dependence networks (e.g., varying depth or alternative regularization methods) would enhance the approach's robustness and generalizability. This could also assist researchers in reproducing the results and applying the technique to different data types, such as NLP or multimodal datasets, thereby expanding its applicability.

Real-World Applicability and Performance on Complex Data:

Weakness: The suggested method shows effectiveness on synthetic datasets (Clevr-4) and obtains satisfactory results on ImageNet; however, it underperforms compared to state-of-the-art methods such as BYOL (Grill et al., 2020) and SimCLR (Chen et al., 2020). This disparity indicates that the adversarial approach might struggle with the complexities and dependencies found in large-scale, real-world data.

Suggestion: Investigating further strategies to boost performance on intricate datasets could enhance the results. For example, integrating adversarial decorrelation loss with sophisticated augmentation techniques similar to those in SimCLR and BYOL may enhance robustness. Moreover, conducting additional experiments in real-world settings, like object detection and segmentation, could demonstrate wider effectiveness and versatility.


Evaluation Metrics for Decorrelation and Embedding Quality:

Weakness: The evaluation primarily depends on Pearson and distance correlation metrics to assess decorrelation. While these metrics are helpful, they may not fully capture the benefits of decorrelation for subsequent tasks or its effect on information retention in embeddings.

Recommendation: Additional metrics like mutual information or latent feature diversity could strengthen the evaluation process. Investigating the impact of decorrelation on particular downstream tasks, such as clustering or retrieval performance, may effectively demonstrate the practical advantages of the proposed approach. For example, assessing the accuracy of k-nearest neighbors on embeddings can provide insights into feature separability, enhancing correlation-based metrics.

**Questions:**

Comparative Benchmarks and Results:
Question: Could you clarify why specific recent methods in decorrelation, such as Barlow Twins, VICReg, and Whitening-MSE, were not included in your benchmarking? Each of these methods has been widely recognized in self-supervised learning for decorrelation, and understanding their comparative performance would strengthen your results.
Suggestion: If possible, compare your approach with these methods to highlight its advantages and limitations in a broader context. Comparing performance on high-complexity datasets like ImageNet would be particularly insightful and would help readers understand its practical strengths.

Impact of Architecture Choices and Hyperparameters:
Question: How sensitive is your approach to the architecture of the dependence networks (e.g., depth, regularization techniques)? Were different architectures tested? If so, how did they impact stability or convergence?
Suggestion: An ablation study or sensitivity analysis on architectural choices (e.g., single-layer vs. multi-layer dependence networks) could enhance the understanding of how robust the method is to architectural variations. This would also assist practitioners in replicating and adapting the method to different data types.

Controlled Redundancy and Task-Specific Tuning:
Question: You introduce a valuable addition: a mechanism to control redundancy levels within the embeddings. Could you explain how users might determine the optimal redundancy level for a given task? Are there any general principles or heuristics that could guide this tuning?
Suggestion: For readers interested in applying this method to various domains, providing guidelines or example scenarios where different redundancy levels improve specific tasks (e.g., classification vs. retrieval) would be helpful. A discussion on dynamically adjusting redundancy levels within the model could also be valuable.

Evaluation Metrics Beyond Correlation:
Question: Why were Pearson and distance correlation metrics primarily chosen for evaluating decorrelation? While informative, they may not fully capture how effective the embeddings are in downstream tasks.
Suggestion: Consider including additional metrics, such as mutual information or clustering performance, which could offer a richer perspective on embedding quality. Including k-nearest neighbor (kNN) accuracy on frozen embeddings or an evaluation of separability in embedding space might better showcase downstream utility, particularly for classification or retrieval applications.

Application Beyond Image Data:
Question: Your experiments focus on image data (e.g., Clevr-4 and ImageNet), which are common in self-supervised learning. Have you considered testing this decorrelation approach in other domains, such as text or multimodal data, where redundancy can also impact generalization?
Suggestion: Extending experiments to other domains would illustrate the versatility and broader relevance of your approach. If this is feasible, showcasing results on text embeddings (e.g., BERT representations) or multimodal embeddings would strengthen the argument for general-purpose decorrelation.

---

### Note · Authors · 2024-11-26

**Comment:**

Dear reviewers,

First and foremost, thank you for your overall detailed and thoughtful reviews! We deeply appreciate the time and effort you dedicated to evaluating our work, your recognition of its potential applications, and your insightful suggestions for improvement.

After careful consideration, we have decided to withdraw the paper and take the time to improve it further by including experiments demonstrating when nonlinear mutual independence outperforms linear decorrelation methods, clarifying how our formulation leads to information maximization, and given the extensive list of suggested additional experiments (i.e., evaluation of our method on additional benchmarks, architectures, and tasks)...

We are particularly grateful to reviewer *1pgN* for pointing out crucial limitations in the current manuscript’s version and the need for a stronger justification of our approach.

Thank you once again for your constructive feedback.

Best regards,

The authors from submission #7756

**Withdrawal Confirmation:**

I have read and agree with the venue's withdrawal policy on behalf of myself and my co-authors.